# Short-Term Dynamics of Vegetation Diversity and Aboveground Biomass of *Picea abies* (L.) H. Karst. Forests after Heavy Windstorm Disturbance

František Máliš [1,2]🆔, Bohdan Konôpka [2,3], Vladimír Šebeň [2,*]🆔, Jozef Pajtík [2] and Katarína Merganičová [3,4]

1  Forestry Faculty, Technical University Zvolen, SK 960 01 Zvolen, Slovakia; malis@tuzvo.sk
2  Forest Research Institute Zvolen, National Forest Centre, SK 960 01 Zvolen, Slovakia;
   bohdan.konopka@nlcsk.org (B.K.); jozef.pajtik@nlcsk.org (J.P.)
3  Faculty of Forestry and Wood Sciences, Czech University of Life Sciences Prague,
   CZ 165 000 Prague, Czech Republic; katarina.merganicova@forim.sk
4  Department of Biodiversity of Ecosystems and Landscape, Institute of Landscape Ecology, Slovak Academy of Sciences, SK 949 01 Nitra, Slovakia
*  Correspondence: vladimir.seben@nlcsk.org; Tel.: +421-455-314-181

**Abstract:** Although forest disturbances have become more frequent and severe due to ongoing climate change, our understanding of post-disturbance development of vegetation and tree–herb layer interactions remains limited. An extreme windstorm, which occurred on 19 November 2004, destroyed *Picea abies* (L.) H. Karst dominated forests in the High Tatra Mts. Here, we studied short-term changes in diversity, species composition, and aboveground biomass of trees and herb layer vegetation, including mutual relationships that elucidate tree–herb interactions during post-disturbance succession. Assessment of species composition and tree biomass measurements were performed at 50 sample plots (4 × 4 m) along two transects 12, 14, and 16 years after the forest destruction. Heights and stem base diameters of about 730 trees were measured and subsequently used for the calculation of aboveground tree biomass using species-specific allometric relationships. Aboveground biomass of herb layer was quantified at 300 subplots (20 × 20 cm) by destructive sampling. Species richness and spatial vegetation heterogeneity did not significantly change, and species composition exhibited small changes in accordance with expected successional trajectories. While aboveground tree biomass increased by about 190%, biomass of annual herb shoots decreased by about 68% and biomass of perennial herb shoots was stable during the studied period. The contribution of trees to total aboveground biomass increased from 83% to 97%. After 16 years of forest stands recovery, tree biomass represented approximately 13% of forest biomass before the disturbance. Herb layer biomass, particularly the biomass of annual herb shoots, was more closely related to tree cover than to tree biomass and its decline could be assigned to gradual tree growth. Our study provides clear evidence that short-term successional processes in post-disturbance vegetation are much better detectable by biomass than by diversity or compositional measures and emphasized the importance of light conditions in tree–herb competitive interactions.

**Keywords:** vegetation succession; species composition; tree cover; herb layer vegetation; tree–herb interactions; carbon sequestration; light conditions

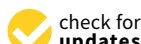



## 1. Introduction

Forest disturbances are important natural drivers of forest ecosystem dynamics and modulate their structure and functioning [1,2]. However, disturbance regimes have been changing due to ongoing climate change and temperate forests are currently affected by large and heavy stand-replacing events around the globe that considerably affect their functioning [3–5]. In Europe, natural disturbances are temporally synchronized [6] and their current severity in *Picea abies* (L.) H. Karst forests is related not only to climate change, but also to disturbance legacies. Large-scale disturbances were present also in the past [7]

and their severity influences vulnerability to current disturbances since structure of spruce forests developed after heavy disturbance is usually even-aged, dense, spatially homogenous, and consequently vulnerable to wind [8,9]. This is also the case of spruce dominated forests in the High Tatra Mts. (Western Carpathians, Central Europe), where heavy wind disturbances driven by specific climate and topographical conditions and consequent bark-beetle outbreaks occur in periodical cycles [10].

In general, disturbed treeless areas are reforested either by planting or by natural regeneration, however, it usually takes a couple of vegetation periods. During the early post-disturbance phase, ground surface is firstly overgrown by herbaceous species which rapidly colonize open habitats e.g., [11]. Later, many of them are partly or fully outcompeted by trees. Although this post-disturbance development of vegetation is very frequent and important for the forest stand future, our knowledge about this period in forest development remains limited [12,13]. Disturbance events disrupt the stable state of forest ecosystems and trigger significant changes in their biodiversity and nutrient cycling. Early post-disturbance stages are characterized by rapid increase of plant diversity driven by eliminated competition of adult trees, newly created habitats, such as bare soil in pits and mounds due to uprooted trees, and overall microsite complexity [12,14]. Many annual, ruderal, and non-forest species may occur at that phase, leading to elevated plant species richness and spatial heterogeneity of vegetation compared to mature stands. In temperate forests, plant species richness reaches the highest levels a few years after a disturbance and then continuously decreases [15,16]. In later successional stages, trees overgrow herbs and take an advantage in competition for resources. In general, interactions between trees and herb layer vegetation are central to successional ecology of forests because they significantly affect the future forest structure and composition [17]. Factors affecting these interactions are various, including the supply of nutrients, light, or water. The canopy openness is among the most important characteristics because it defines light conditions which are essential for forest understory [18,19]. However, light-defining measures may not be sufficiently sensitive to study short-term successional processes or even tree–herb competition for other resources, such as water or nutrients. The evaluation of plant biomass divided to functional groups (e.g., herbs and trees) should be a more sensitive measure to better elucidate post-disturbance dynamics and competitive relationships. Moreover, biomass quantification enables the assessment of carbon sequestration as another essential forest ecosystem feature heavily affected by disturbance events [20]. Since carbon storing by forests is important for mitigation of climate change, fast forest restoration and reaching full canopy are required to compensate for carbon losses caused by previous forest stand destructions [21]. On the other hand, both gradual canopy closing, and competitive pressure of trees suppress herb layer diversity, creating a management trade-off between carbon storage and biodiversity [22,23].

This paper deals with short-term changes (over a period of 2 to 4 years) of tree–herb layer interactions in post-disturbance vegetation more than a decade after the large windthrow event in the High Tatra Mts. Species composition and plant biomass were inventoried at 50 permanent plots situated within two linear transects 12, 14, and 16 years after the disturbance. By analyzing these data, we aim to address the questions (i) whether plant species richness, composition, spatial heterogeneity and herb layer biomass of post-disturbance vegetation decreased over the studied period, and (ii) whether and how the tree growth affected these temporal changes of the herb layer.

## 2. Materials and Methods

### 2.1. Study Area

Our study focused on the High Tatra Mts., which form a part of the Tatra National Park situated in northern Slovakia. The bedrock is typically formed by fluvioglacial sediments of granodiorites. Forest soils are mostly represented by lithic leptosols and podzols. The climate is cold and humid with annual mean temperature of about 5.0 °C, and annual precipitation totals of over 1000 mm. Snow cover lasts between 110–130 days [24].

A substantial part of forests in this area was damaged by an extreme windthrow event on 19 November 2004 (Figure 1). The wind destroyed forests dominated by *Picea abies* at altitudes from 700 to 1400 m a.s.l. The disturbed area was concentrated in a 35 km long continuous belt stretching in a west-east direction (Figure 1). Nearly all forest stands inside this belt were entirely destroyed (more trees were uprooted than stem-broken), except for a few tree clusters with high shares of *Larix decidua* (Mill.) and/or *Pinus sylvestris* (L.). The damaged forests covered an area of nearly 130 km$^2$ [25] and considerably affected forest ecosystem services [26].

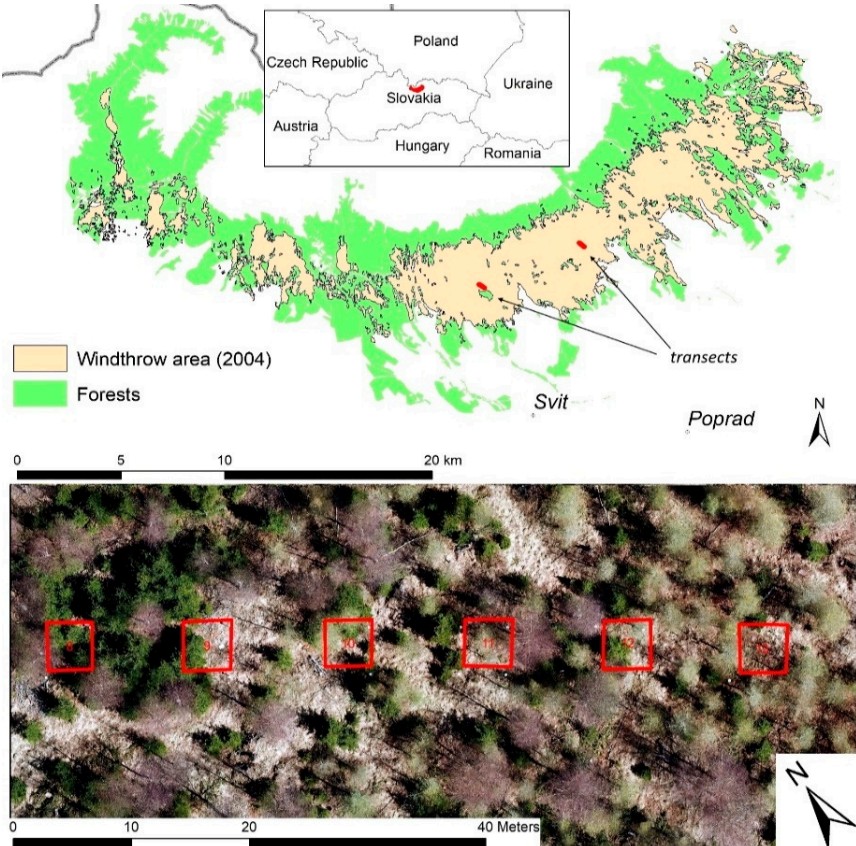

**Figure 1.** Study area situated in the High Tatra Mts. (the Western Carpathians) affected by the windthrow event in 2004. Two transects are located in salvage-logged stands with combined natural and artificial tree regeneration dominated by *Picea abies* (L.) H. Karst, *Larix decidua* (Mill.), and *Betula pendula* (L.). Each transect comprises 25 plots of 16 m$^2$ in size (4 × 4 m), separated by 8 m gaps. The areal picture was snapped from a drone in the early spring of 2018 and shows the middle part (plots 8–13) of the DD transect.

To study diversity and aboveground biomass of post-disturbance vegetation, we established two research transects—one near the site called "Danielov dom" (DD) and another one close to the "Horný Smokovec" (HS) village. Since both sites belong to the territory with the lowest degree of nature protection within the national park, the fallen trees were salvage-logged and combined reforestation was applied using both natural regeneration and planting of various tree species (mostly *Picea abies*, *Pinus sylvestris*, *Larix decidua*) in spatially and temporally variable schemes to increase the structural complexity of future stands [27]. Due to this specific reforestation practice, tree density and species composition at and along the transects vary from completely treeless sites to dense young stands.

The transects are located approximately in the center of the disturbed belt (Figure 1). The orientation of transects is from northwest to southeast. The altitude of transects varies between 970–1000 m a.s.l. (DD transect) and between 920 and 950 m a.s.l. (HS transect). Each transect comprises 25 square plots of 16 m² in size (4 × 4 m). The plots are situated 8 m apart to ensure their spatial independence. Thus, the total length of each transect is 292 m. We fixed the corners of the plots with wooden pegs, and labeled all young trees occurring within the plots and higher than 10 cm with a code written on a metallic tag.

### 2.2. Data Sampling

For the calculation of tree biomass, we recorded taxonomic identity (species), tree height, and diameter at stem base ($d_0$ hereinafter) of each labeled tree at the plot. We performed tree measurements in the second half of the growing seasons of 2016, 2018, and 2020 (always after the annual diameter growth was finished). Altogether, between 673 and 743 individual trees were measured at both transects in individual years (Table 1).

**Table 1.** Main characteristics of the measured trees at the DD and HS transects in the years 2016, 2018, and 2020.

| Tree Characteristics | Transect | Year 2016 | Year 2018 | Year 2020 |
|---|---|---|---|---|
| Number of measured trees | DD | 354 | 357 | 367 |
| | HS | 319 | 350 | 376 |
| Average tree diameter at stem base (Standard deviation) /mm/ | DD | 31.7 (26.2) | 41.0 (34.0) | 51.8 (43.0) |
| | HS | 25.3 (30.0) | 31.8 (38.3) | 38.5 (46.0) |
| Lorey's mean height /m/ | DD | 3.8 | 4.9 | 6.0 |
| | HS | 5.0 | 5.9 | 7.1 |

To sample species composition of vegetation, we inspected each 4 × 4 m plot for present vascular plants and for each taxon, including tree species, we estimated their cover values in percentages. Species nomenclature follows the list [28]. We assessed the cover of tree species regardless of vertical structure, i.e., we did not divide trees into separate vertical layers. To sample the aboveground herb layer biomass, we harvested all vascular plants excluding trees from six square subplots of 20 × 20 cm systematically distributed within each plot to better represent spatial variability of herb layer biomass. We applied the shoot approach, i.e., we collected all plants' shoots present within the subplots. Next, we merged six subsamples together and separated the collected biomass to two lifespan groups, annual and perennial herb shoots. Biomass of annual herb shoots (BAHS) comprised those aboveground plant parts, which live only a single vegetation period (one year), for example, the entire aboveground body of *Calamagrostis villosa* Chaix ex Vill.) J. F. Gmelor leaves of *Vaccinium myrtillus* (L.). Biomass of perennial herb shoots (BPHS) comprises those aboveground plant parts, which live more than one year, usually several years, for example, the entire body of *Vaccinium vitis-idaea* (L.) or stems of *Vaccinium myrtillus* (L.). We assessed species composition and sampled plant biomass three times, in 2016, 2018, and 2020, during the growing peak of the vegetation period (mid-August). In 2018 and 2020, we systematically shifted locations of biomass subplots by 0.5 m compared to previous years sampling.

### 2.3. Data Analyses

We estimated the aboveground tree biomass from tree measurements (tree height and diameter at stem base (DAB)) using published species-specific allometric relations (Table 2) defined by Equation (1). We calculated the biomass stock separately for each transect, first at a plot level (as a sum of all trees at the plot), then as a mean value from 25 plots:

$$W_i = e^{(b_0 + b_1 . \ln DAB + b_2 . \ln h)} . \lambda \qquad (1)$$

where $W_i$ is aboveground biomass (kg), *DAB* is diameter at stem base (mm), *h* is tree height (m), $b_0$, $b_1$, $b_2$ are species-specific parameters (Table 2), $\lambda$ is the transformation correction factor (Table 2).

**Table 2.** Parameters and correction factors for allometric relationships used to estimate aboveground tree biomass based on diameter at stem base and tree height as independent variables (see Equation (1)).

| Tree Species | Parameter | | | $\lambda$ * | Source—Reference |
|:---:|:---:|:---:|:---:|:---:|:---:|
| | $b_0$ | $b_1$ | $b_2$ | | |
| *Betula pendula* (L.) | −1.545 | 2.032 | 0.586 | 1.033 | own unpublished model ** |
| *Larix decidua* (Mill.) | −0.942 | 2.022 | 0.543 | 1.067 | [29] |
| *Salix caprea* (L.) | −0.912 | 1.801 | 0.854 | 1.021 | [29] |
| *Picea abies* (L.) H. Karst | −0.696 | 2.002 | 0.4 | 1.037 | [30] |
| *Sorbus aucuparia* (L.) | −1.666 | 2.149 | 0.47 | 1.015 | [29] |
| *Pinus sylvestris* (L.) | −0.466 | 1.856 | 0.476 | 1.031 | [31] |
| Other deciduous *** | −1.692 | 2.145 | 0.533 | 1.036 | [32] |

* $\lambda$—correction factor (see for instance [33]). ** The model was derived from the data introduced in the paper by Konôpka et al. [34] *** Other deciduous tree species were calculated using the allometric relationship derived for *Populus tremula*.

To evaluate temporal changes in vegetation composition, we compared occurrence frequency and cover of each species between sampling periods and separately for transects. We calculated the frequency as a number of plots occupied by a particular species out of all 25 plots at the transect and expressed it as a percentage. We calculated the cover as a mean of cover values from plots where the particular species occurred. To assess the compositional shift from early post-disturbance vegetation of open habitats to species composition of closed mature spruce forests, we evaluated the proportion of species that are considered as diagnostic for local forest vegetation, i.e., *Vaccinio-Piceetea* class [35]. We calculated their proportion for each plot as the number of present *Vaccinio-Piceetea* species divided by species richness.

We expressed mean vegetation heterogeneity at one transect with Bray-Curtis dissimilarity between 4 × 4 m plots of the transect calculated using the *vegdist* function in the *vegan* R package [36] in R [37]. We transformed species cover values with the square root transformation using *sqrt* function. We tested temporal changes in species richness, proportion of *Vaccinio-Piceetea* species and biomass as a pairwise comparison with the Wilcoxon rank sum test with Bonferroni correction using *wilcox.test* function. To test the change in spatial heterogeneity, we compared plot dispersions around multivariate centroids between the samplings in 2016, 2018, and 2020 [38]. We tested the statistical significance of the differences between the 2016, 2018, and 2020 samplings based on the multivariate homogeneity of group dispersions [39]. We applied 999 permutations restricted to samples paired over two sampling periods. We produced scatter and bar plots using the *ggplot2* package [40].

To reveal whether tree biomass or sum of tree cover is a better predictor of herb layer vegetation biomass, we compared the values of coefficients of determination of linear regression models fitted to values of herb layer biomass, tree biomass, and sum of tree cover within herb biomass groups and years separately (Table 3). We calculated the sum of tree cover as a sum of cover values of each tree species present in the plot, thus the final value could be higher than 100%.

**Table 3.** Mean species richness, vegetation spatial heterogeneity (expressed by Bray-Curtis dissimilarity), aboveground biomass divided into three plant groups (biomass of annual herb shoots, biomass of perennial herb shoots, tree biomass), and tree layer cover at 16 m$^2$ plots within transects and sampling periods. Statistical significance of differences between sampling periods was tested as a pairwise comparison using the Wilcoxon rank sum test with Bonferroni correction and is indicated by letters in brackets (A, B, C) at *p* level < 0.05. Different letters indicate significant differences between periods, while the occurrence of the same letter suggests the insignificant difference.

| Diversity Characteristics of Vegetation | Transect | Year 2016 | Year 2018 | Year 2020 |
|---|---|---|---|---|
| Mean species richness per plot (number of tree species out of all species) | DD | 16.8 (2.4) (A) | 16.8 (2.6) (A) | 17.5 (3.0) (A) |
| | HS | 16.4 (3.2) (A) | 16.1 (3.3) (A) | 16.5 (3.4) (A) |
| Vegetation beta diversity (spatial heterogeneity) | DD | 0.484 (A) | 0.477 (A) | 0.474 (A) |
| | HS | 0.446 (A) | 0.448 (A) | 0.443 (A) |
| Mean biomass of annual herb shoots /kg m$^{-2}$/ | DD | 292.7 (A) | 187.0 (B) | 72.2 (C) |
| | HS | 263.8 (A) | 164.4 (B) | 99.9 (C) |
| Mean biomass of perennial herb shoots /kg m$^{-2}$/ | DD | 80.4 (A) | 35.7 (AB) | 15.8 (B) |
| | HS | 37.2 (A) | 40.8 (A) | 15.0 (A) |
| Mean tree biomass /kg m$^{-2}$/ | DD | 1247.1 (A) | 2231.8 (AB) | 3552.5 (B) |
| | HS | 1069.0 (A) | 1982.7 (B) | 3178.8 (B) |
| Mean sum of tree cover /%/ | DD | 30.2 (A) | 45.6 (B) | 55.4 (B) |
| | HS | 28.3 (A) | 43.3 (B) | 50.4 (B) |

## 3. Results

Mean species richness recorded per 16 m$^2$ plot was 17 (min. 9, max. 26) for either of transects. On average, three species out of the total 17 were tree taxa (min. 0, max. 8) (Table 3). The DD transect was dominated by coniferous tree species, mainly *Picea abies* and *Larix decidua*, which are typical for mature stands, while at the HS transect, deciduous and early successional tree species prevailed, mainly *Betula pendula*. (Supplementary Tables S1 and S2, Figure 2A). During the studied period, species richness was stable with only slight and insignificant increase at the DD transect. In addition, the proportion of species considered diagnostic for *Vaccinio-Piceetea* class did not significantly differ between sampling periods. Similarly, we did not find any significant changes in vegetation heterogeneity over time (Table 3). Herb layer was dominated particularly by grasses with annual aboveground biomass (*Calamagrostis villosa* (Chaix ex Vill.) J. F. Gmel, *C. arundinacea* (L.) Roth, *Avenella flexuosa* (L.) Parl.). Plants with longer life span of biomass were also frequent, particularly *Vaccinium myrtillus* (L.) (Supplementary Tables S1 and S2). Temporal changes in frequency or cover of these herb layer dominants were inconsistent in terms of some functional groups. However, several early-successional species, non-forest taxa, or those which have suitable conditions on forest clearings, such as *Epilobium angustifolium* (Lam.), *Rubus idaeus* (L.), *Phleum pratense* (L.), *Galeopsis bifida* (Boenn.), *Sambucus racemose* (L.), *Calluna vulgaris* (L.) Hull., *Calamagrostis villosa* (or *Avenella flexuosa* (L.) Parl., decreased in frequency or cover. On the other hand, the occurrence of *Vaccinium myrtillus* (L.), a typical dominant taxon of mature mountain spruce forests, increased.

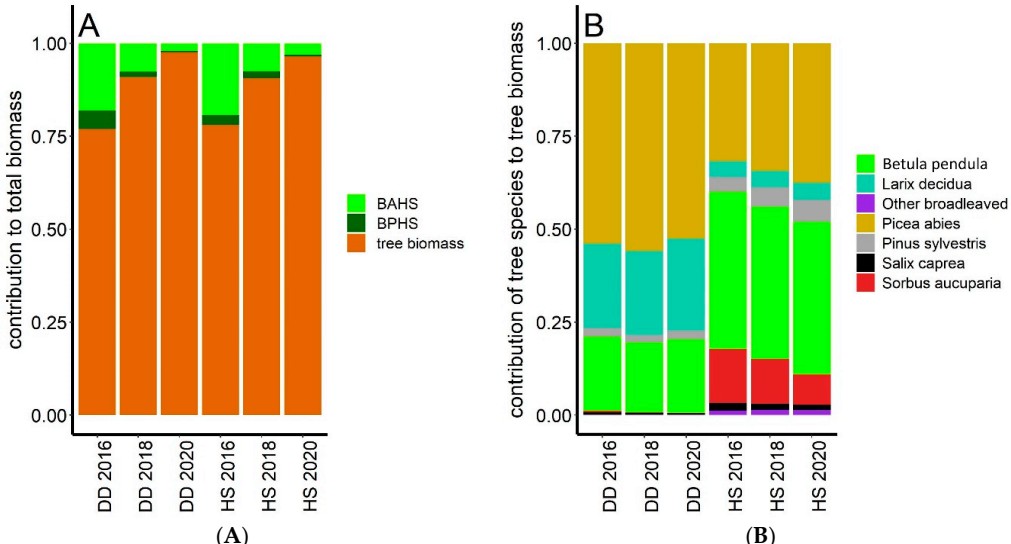

**Figure 2.** Relative contribution of biomass of annual herb shoots (BAHS), biomass of perennial herb shoots (BPHS), and tree biomass to total biomass (**A**) and tree species to tree biomass (**B**) in the DD and HS transects in 2016, 2018, and 2020. Values and statistical significance of temporal differences are shown in Table 3.

Likewise, tree species composition did not significantly change over the studied period. The frequency and cover of several tree species, particularly *Betula pendula* (L.), *Picea abies* (L.) H. Karst, *Larix decidua* (Mill.), or *Pinus sylvestris* (L.) substantially increased, which resulted also in the slight increase of mean number of trees per plot (Table 3, Tables S1 and S2). However, this did not result from the emergence of new recruits, but rather from tree growth, since the presence of trees at a plot was estimated using a shoot approach. This means that individual plants identified within a plot did not have to be rooted inside the plot, instead the trees growing outside the plot simply enlarged their crowns that grew into the plot.

Contrary to negligible changes in species composition over the studied period, biomass of tree and herb layers considerably changed (Table 2, Figure 2). Tree biomass between 2016 and 2018 increased at both transects by about 82% (calculated as ((biomass in 2018−biomass in 2016)/biomass in 2016) × 100) and between 2018 and 2020 by about 60% on average. Over the entire period of 4 years, tree biomass increased by about 190%. In the case of herb layer, BAHS significantly decreased at both transects by about 68% over the 4-year-long studied period, while BPHS was more stable, especially at HS transect (Figure 2A, Table 3). BAHS was much higher compared to the BPHS, but tree biomass largely dominated in the entire ecosystem plant biomass (Figure 2A). Over the studied period, tree dominance further increased and its contribution to the entire ecosystem plant biomass increased from 82%–84% to 97%. Comparison of tree species contribution to tree biomass (Figure 2B) showed that the proportion of coniferous tree species typical for mature stands, i.e., *Picea* abies (L.) H. Karst and *Larix decidua* (Mill.), increased at the HS transect at the expense of pioneer tree species, such as *Betula pendula* (L.), *Sorbus aucuparia* (L.), or *Salix caprea* (L.).

Coefficients of determination of linear regressions describing the relationship between the herb layer biomass and tree biomass or tree cover revealed that tree cover explained more of the variation in herb layer biomass than tree biomass (Table 4). Therefore, we consequently analyzed the relationship between the herb and tree layers using tree cover as a primary predictor. While BPHS was not substantially affected by tree cover, we found a clear negative correlation between tree cover and BAHS. Additionally, we found that the relationship between BAHS and tree cover substantially weakened in time (Figure 3, Table 4). This pattern was more evident for DD transect. The number of all plant species present at the plot was much less related to tree cover (Figure 4) and the relationship did not change over time, indicating that plant species were able to persist despite tree growth.

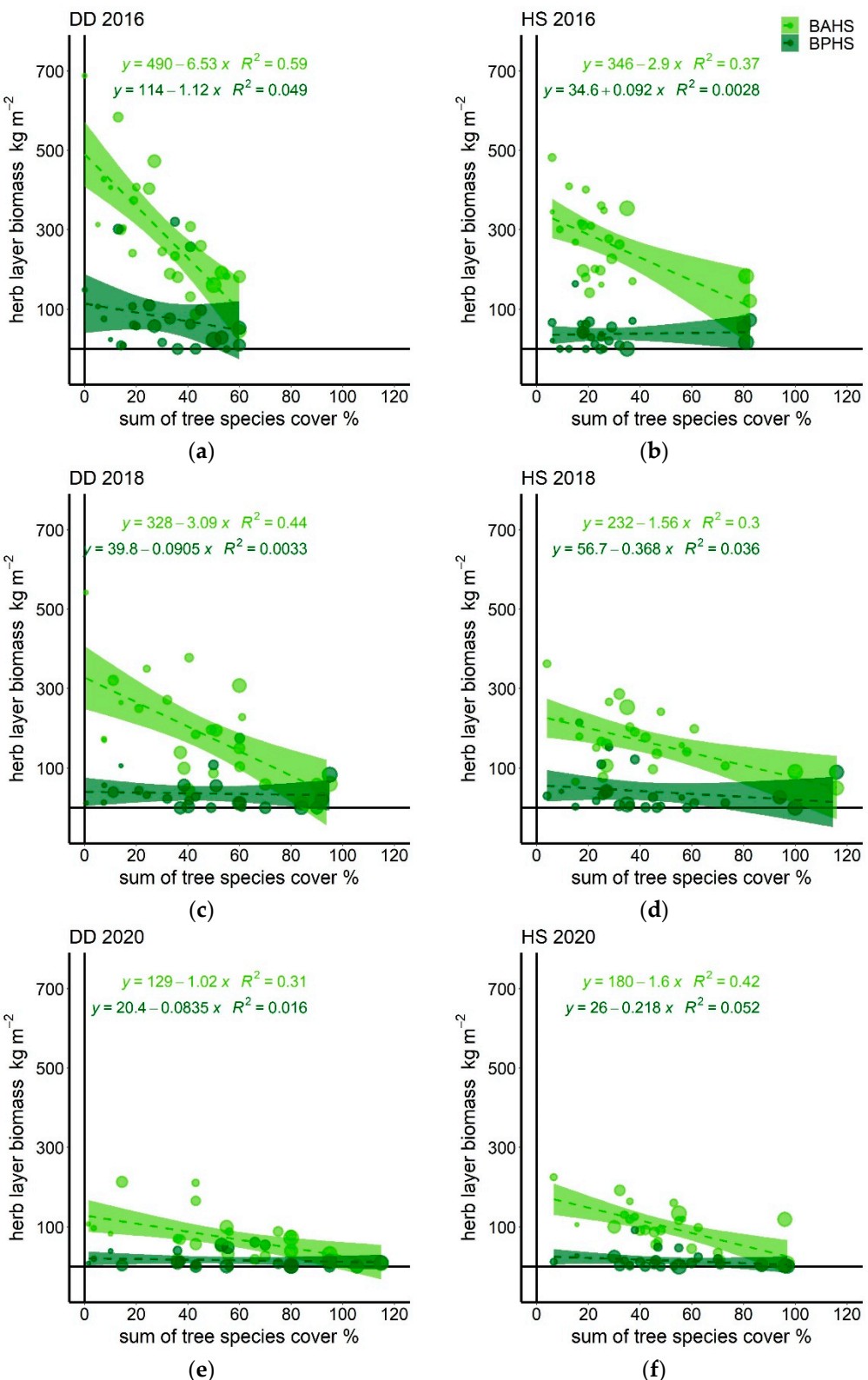

**Figure 3.** Relationship between herb layer biomass (all species except for trees) and sum of tree species cover at transects (DD and HS) within sampling periods expressed by linear regressions (respective equations are presented at the top of each graph, where *x* is the sum of tree species cover, and *y* is the herb layer. Explanations: (**a**) DD, Year 2016, (**b**) HS, Year 2016, (**c**) DD, Year 2018, (**d**) HS, Year 2018, (**e**) DD, Year 2020, (**f**) HS, Year 2020.

**Table 4.** Coefficients of determination of linear regression models describing the relationships between herb layer biomass (biomass of annual herb shoots (BAHS), biomass of perennial herb shoots (BPHS), and both together) and tree biomass or tree cover.

| Relationship | Herb Layer Biomass Group | $R^2$ Values | | | | | |
| --- | --- | --- | --- | --- | --- | --- | --- |
| | | Year 2016 | | Year 2018 | | Year 2020 | |
| | | DD | HS | DD | HS | DD | HS |
| Herb layer biomass vs. tree biomass | BAHS | 0.191 * | 0.158 | 0.343 * | 0.104 | 0.185 * | 0.064 |
| | BPHS | 0.025 | 0.005 | 0.011 | 0.021 | 0.014 | 0.101 |
| | Both | 0.059 | 0.024 | 0.105 | 0.033 | 0.068 | 0.029 |
| Herb layer biomass vs. tree cover | BAHS | 0.592 * | 0.374 * | 0.440 * | 0.301 * | 0.314 * | 0.421 * |
| | BPHS | 0.049 | 0.003 | 0.003 | 0.036 | 0.016 | 0.052 |
| | Both | 0.171 * | 0.047 | 0.127 * | 0.085 * | 0.111 * | 0.112 * |

Asterisks (*) indicate statistical significance of the model based on *p*-value < 0.05.

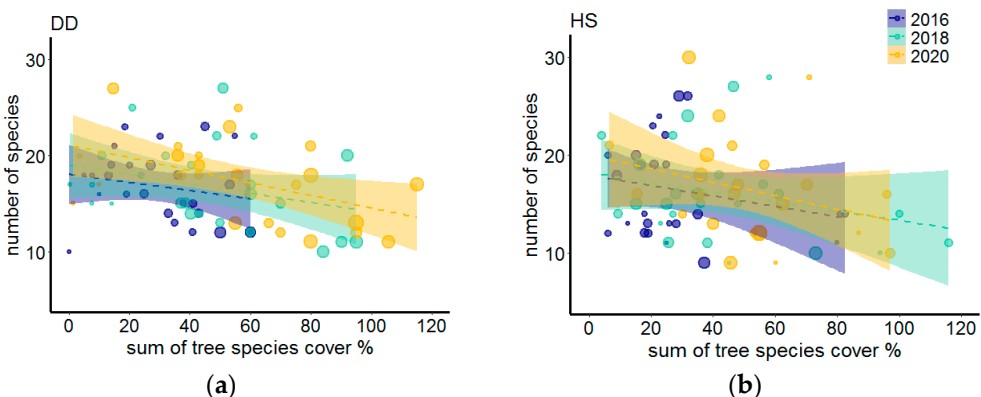

(**a**)  (**b**)

**Figure 4.** Relationship between number of all plant species recorded at 4 × 4 m plots and sum of tree species cover at the transects (DD and HS) in sampling periods expressed by linear regressions. Point size is scaled by tree biomass. Different colors are used for sampling periods. Shaded areas represent 95% confidence intervals around regression models. Explanations: (**a**) transect DD, (**b**) transect HS.

## 4. Discussion

### 4.1. Status and Temporal Changes in Post-Disturbance Vegetation

At both transects, the tree species of locally late-successional mature stands, i.e., *Picea abies* (L.) H. Karst or *Larix decidua* (Mill.) [10,41], prevailed in tree regeneration accompanied by several pioneer tree species, most frequently *Betula pendula* (L.). This tree species composition was also found in other studies and regions with post-disturbance vegetation of mountain spruce forests e.g., [42,43]. Rather high tree diversity is primarily the result of natural regeneration [44,45], since except for *Picea abies* (L.) H. Karst, most frequent trees were deciduous pioneer species that were not planted. The number of tree species increased only slightly, likely due to the tree growth as the shoot approach was applied during sampling. In the post-disturbance area of the High Tatras, Konôpka et al. [46] observed an increase in tree species diversity between the third and the seventh year after the windthrow, while later, the number of tree species was stabilized. Cover and frequency of almost all tree species increased regardless of their affinity to specific successional phases, indicating continuous development of forests without strong competition in tree layer. However, temporal changes in contribution of tree species to entire tree biomass indicated that coniferous tree species typical for late-successional mature stands increased their prevalence, thus taking an advantage of their competition abilities already in this early successional stage of forest stand development [17]. However, their contribution to tree biomass has not reached the pre-disturbance state yet. In 1996, as much as 98% of tree

carbon stock was stored in three most common coniferous species (*Picea abies*, *Larix decidua*, *Pinus sylvestris*, [46]), while in our case, they contributed by less than 75 or 50% depending on the transect (Figure 2). Considering the current tree species composition and the longevity of individual tree species, we may expect that the biomass contribution of the three coniferous species will further increase because the other co-occurring species are short-living ones with early growth culmination [47–49]. Hence, future stands are likely to approach the pre-disturbance tree species composition.

Mean number of all species per plot, including tree species, did not change significantly over the studied period. Species richness of post-disturbance vegetation usually rapidly increases after a disturbance event and later when competitive species take place it gradually declines [12,15]. Investigated vegetation represents plant communities in developmental stages 12–14–16 years after the disturbance, which is very likely already after the peak of vegetation richness [16]. At the post-disturbance diversity peak, plant communities are saturated by many non-woody, early-successional, ruderal, or simply those species which benefit from open habitats of forest clearings with high levels of light and disturbed soil [11,42]. Some of those were still present at our sites, for example, *Epilobium angustifolium* (L.) or *Calluna vulgaris* (L.), but their frequency and cover decreased over the studied period. Additionally, the frequency or cover of other plants, mostly grasses, decreased, e.g., *Avenella flexuosa*, *Calamagrostis villosa*, *Luzula luzuloides (*Lam.) Dandy et Wilmott. These species occur also in mature and old-growth stands but with low cover values e.g., [41,50], while after canopy opening, their abundance rapidly increases [42,51]. The persistence of species that are typical for mature late-successional forests in post-disturbance vegetation was confirmed also by the high and stable proportion of *Vaccinio-Piceetea* class species during sampling periods.

Along with advancing succession, not only the number of plant species, but also plant spatial heterogeneity [12] is expected to decrease. However, the period of four years our study is based on is likely too short to observe any significant changes. Perhaps, this might be related to lower levels of spatial heterogeneity [52] and also faster homogenization of young forest structure [53] at salvage-logged sites as this type of post-disturbance management was applied at our transects. Salvage logging certainly favored also high abundance of grasses. Several studies identified higher species richness and cover of graminoids at salvage-logged compared to non-intervention sites which likely triggered also different successional trajectories of post-disturbance herb layer vegetation [11,51,52]. Decline of grasses identified in our study possibly induces a compositional convergence of salvage-logged and non-intervention sites, however, our study did not examine successional processes following different management approaches, thus further research is needed to verify this assumption.

*4.2. Impact of Trees on Herbaceous Vegetation*

We found that the species richness and vegetation heterogeneity did not significantly change over the observed period, while herb layer biomass did (Figures 3 and 4). Interannual changes in biomass could be assigned to variability of climatic conditions. Indeed, the weather during the vegetation period in 2016 was more humid and colder compared to 2018 and 2020 and the vegetation period in 2018 was the warmest [54]. However, the reduction of the observed herb layer biomass is more likely the result of successional processes rather than weather differences, because low temperature is the limiting factor of plant growth in studied mountain spruce forests, not water deficit [55]. We further tried to elucidate what affected herb layer biomass more, whether tree layer biomass or tree cover. The results suggested that tree cover was much more important, but only when predicting BAHS. This indicates that plants with annual biomass need sufficient light conditions to build their bodies and when this essential source is limited [18], their biomass rapidly declines [56]. BPHS was very weakly correlated to tree biomass or cover. Moreover, the impact of tree cover on BAHS and also the amount of BAHS itself considerably and gradually decreased during four years of monitoring, suggesting that the ongoing succession and tree

growth suppress herb layer biomass. Grass species, particularly *Calamagrosis villosa*, *C. arundinacea*, or *Avenella flexuosa*, contributed to BAHS most. As mentioned above, they also occur in mature stands but with much lower abundance. These grassy species certainly benefited from higher light levels after disturbance, e.g., [11,22]. Thus, identified decrease of BAHS along with the increasing tree cover and continuing succession could be assigned mostly to grasses. Observed decrease of BAHS also clearly fits to expected post-disturbance successional pathways. Biomass of grasses was approximately 15-times greater in 2010 (six years after the disturbance) than in 2005, the first year after the disturbance [41,57]. Compared to unaffected reference, forest stands grasses had more biomass already in the first year after the disturbance and this difference considerably increased. Similar development was identified also in areas without salvage logging, although in the non-intervention sites, the contribution of BPHS was slightly higher due to higher abundance of *Vaccinium myrtillus*. While these studies reflect the phase of rapid increase of BAHS just after the forest disturbance, our results showed the opposite trend driven by forest stand recovery. Additionally, the relationship between species richness and tree cover did not change over the investigated period, which implies that in this post-disturbance developmental stage, plants still find enough space to persist, however, their growth [biomass] is already limited by the growth of new tree generation.

### 4.3. Forest Growth and Carbon Accumulation

While herb layer biomass decreased, tree biomass almost tripled over four years. This indicates fast growth of young forest stands which is essential for post-disturbance recovery of carbon stock [20,21]. Mean carbon stock of forest stands before the disturbance was 10 g/m$^2$ [46,58]. Currently, 16 years after the disturbance event, the values reached only approximately 13% of original pre-disturbance values. Although the biomass stock (i.e., amount of stored carbon) of young forest stands is small compared to mature forests, their biomass increment [net primary productivity] may be even larger than in old stands, e.g., [59]. Due to this fast accumulation of tree biomass, the overall carbon balance [difference between carbon uptake by photosynthesis and losses by respiration] of forests at the disturbed area became positive already 10 years after the disturbance [60]. Usually trees are considered to contribute to carbon accumulation, but recently, it was shown [61] that non-woody herb layer vegetation can seriously decrease carbon loss via its sequestration in biomass and mitigation of soil respiration due to shadowing ground surface. Particularly high productivity of graminoids, the cover and biomass of which is very high also in the studied area [41], is important from this point of view. Obviously, rapid increase of tree biomass compensates carbon losses due to reduction of herb biomass. At the same time, an increasing amount of carbon fixed in woody compartments of trees in comparison to short-living organs of herb layer vegetation slows down carbon cycling in the ecosystem, and causes gradual carbon accumulation [62]. In general, carbon fluxes require several decades to recover, while aboveground or total tree biomass requires much longer recovery time, usually exceeding 100 years [20,63]. Moreover, recovery time is related to disturbance type and severity, e.g., forests affected by storms require around 40 years to reach the pre-disturbance state [63]. Based on these findings, we assume that carbon stocks and fluxes of the newly established vegetation in the High Tatra Mts. require at least a few decades to recover. Since post-disturbance carbon dynamics depends on the forest type and climate [64], the precise quantification of forest recovery processes in the studied area requires long-term research of tree and herb layer vegetation development and their interactions.

## 5. Conclusions

Our study of post-disturbance vegetation dynamics revealed that although vegetation diversity measures did not significantly change over the short-term period, the aboveground biomass of tree and herb layers, including their relationship, changed considerably. Results correspond to usual successional development in the phase when gradual growth

of new tree generation suppresses herb layer vegetation. The increasing tree cover had a stronger impact on herb layer biomass than tree biomass, which emphasizes the importance of light conditions on herb layer biomass. Our findings suggest that even short-term observations could identify temporal changes in vegetation and elucidate tree–herb layer interactions when sensitive measures such as biomass are used.

Further, our research highlighted the importance of continual post-disturbance monitoring to provide a better understanding of forest stand recovery, which can subsequently help to manage forests in an optimal way to satisfy a great variety of human society demands on forests of protected areas including biodiversity and carbon accumulation.

**Supplementary Materials:** The following are available online at https://www.mdpi.com/1999-4907/12/1/97/s1, Table S1: Species and their frequency (Freq.) and mean cover (Cover) at the DD transect, Table S2: Species and their frequency (Freq.) and mean cover (Cover) at the HS transect.

**Author Contributions:** Conceptualization, F.M. and B.K.; Data Curation, V.Š. and J.P.; Funding Acquisition, B.K.; Investigation, F.M., B.K., V.Š. and J.P.; Methodology, F.M. and B.K.; Visualization, V.Š. and J.P.; Supervision, F.M.; Writing—Original Draft Preparation, F.M., B.K., and K.M.; Writing—Review and Editing, all authors. All authors have read and agreed to the published version of the manuscript.

**Funding:** This research was funded by grant "EVA4.0", No. CZ.02.1.01/0.0/0.0/16_019/0000803 financed by OPRDE, also by the projects APVV-18-0086 and APVV-19-0387 from the Slovak Research and Development Agency and by the project: "Scientific support of climate change adaptation in agriculture and mitigation of soil degradation" (ITMS2014+ 313011W580) supported by the Integrated Infrastructure Operational Programme funded by the ERDF.

**Institutional Review Board Statement:** Not applicable.

**Informed Consent Statement:** Not applicable.

**Acknowledgments:** We are grateful to forest managers from State Forests of High Tatra National Park for their support and Martin Valentík for his help during field work.

**Conflicts of Interest:** The authors declare no conflict of interest.

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
