# Peer review of "Short-Term Dynamics of Vegetation Diversity and Aboveground Biomass of Picea abies (L.) H. Karst. Forests after Heavy Windstorm Disturbance"

_forests, doi:10.3390/f12010097_

Round 1
Reviewer 1 Report
At the beginning I will introduce minor issues:
Line 24.: The area of subplots is extremely low (0.25 x 0.25 cm) and it does not correspond with the dimensions shown in "Data sampling" part (Line 145: 20x20cm).
Line 114: "km2" - there should be a power notation
Figure 2: The position of the plots should be aligned. The legends of plots are truncated. It should be corrected.
Figure 3: The uniform vertical axis scale makes the plots hard to read.
Line 225: The letter (A, B, C) which show differences between sampling periods should be more clearly described.
Lines 239-249: What method (equation) was used to determine the percentage of biomass increase.
More serious issues:
There is no even general description of measured trees of parameters.
In reference 29 (self-citation) authors show equations based on dbh or height. There is no equation based on DAB and height. I suppose that author prepared a new equation based on the same data, but it has not been published yet.
Table3 (Fig. 3 and 4). Are the analyzed relations statistically significant? Lines 265-266: Is this observation based only on plots or do you perform any statistical analysis? If not, it is hard to conclude, that most of the conclusions are supported by research results.
Author Response
At the beginning I will introduce minor issues:
Line 24.: The area of subplots is extremely low (0.25 x 0.25 cm) and it does not correspond with the dimensions shown in "Data sampling" part (Line 145: 20x20cm).
Yes, it was our mistake in the Abstract. The correct size is 20 x 20 cm, thus, it has been corrected in the Abstract.
Line 114: "km2" - there should be a power notation
Thanks,…km2 - corrected.
Figure 2: The position of the plots should be aligned. The legends of plots are truncated. It should be corrected.
Yes, thanks, this problem with the figure occurred after inserting it into the template. It is corrected in the revised version.
Figure 3: The uniform vertical axis scale makes the plots hard to read.
Thanks for the comment. We agree that different scales would make the plots easier to read, but we still prefer our approach with the same scale size because it emphasizes the gradual change of tree-herb layer interactions over post-disturbance succession. Moreover, uniform axis scales better show the decrease in herb layer biomass amount (vertical axis) and increase of tree cover (horizontal axis) over studied periods. Further, the same scale allows readers to clearly see the differences in the trends of the linear models.
Line 225: The letter (A, B, C) which show differences between sampling periods should be more clearly described.
We included additional information about the method used for testing and how letters (A, B, C) indicate significance of differences. Please see the revised caption of Table 3.
Lines 239-249: What method (equation) was used to determine the percentage of biomass increase.
We specified it in the revised version on lines 246-247 as follows "calculated as ((biomass in 2018 – biomass in 2016) / biomass in 2016) * 100”
More serious issues:
There is no even general description of measured trees of parameters.
We have added a table (Table 1) that shows the basic tree characteristics. Originally, this table was included in supplement.
In reference 29 (self-citation) authors show equations based on dbh or height. There is no equation based on DAB and height. I suppose that author prepared a new equation based on the same data, but it has not been published yet.
Yes, we agree and appreciate this comment. Thus, we changed the reference number 29 in Table 1 (currently Table 2) to the statement “own unpublished model” and added the source of data used for modelling.
Table 3 (Fig. 3 and 4). Are the analyzed relations statistically significant? Lines 265-266: Is this observation based only on plots or do you perform any statistical analysis? If not, it is hard to conclude, that most of the conclusions are supported by research results.
Yes, the significance of the linear regressions was not shown or mentioned. In the revised version, the statistical significance of models is indicated by asterisk (*) in the current Table 4 (original Table 3). We reformulated the sentence on lines 265-266 to clarify the message. Primarily we wanted to say that the relationships weakened over the studied period, which could be visible from the trends of the linear models (Fig.3) as well as from regression parameters. Therefore, we have changed the formulation and corrected the meaning (please see the change below) and also added regression equations that include regression parameters to the plots (Fig. 3).
Previous version:
Additionally, we found that the explanatory power of the models describing the relationship between annual biomass and tree cover substantially decreased in time (Figure 3, Table 3).
Current version:
Additionally, we found the relationship between annual biomass and tree cover substantially weakened in time (Figure 3, Table 4).

Reviewer 2 Report
it is necessary to correct the title "Short Term Dynamics of Vegetation Diversity and Aboveground Biomass of Picea abies Forests after Heavy Disturbance", because the authors consider only the consequences of the storm, and not all variants of severe desturbances.
The authors in the methods section use both passive and active voice when presenting the material. It is better to use only active voice.
Latin plant names are not always italicized (see 226, 227, 231 etc.).
There is confusion between annual biomass, biomass of annual species and biomass of annual parts of herbs. Annual biomass for example in the sentence (l 214) means the biomass that has grown over the year, but in the context it is the biomass of annual shoots. Authors should clearly mark "biomass of annual shoots" through the text (for example using an abbreviation - BAS) so that there is no confusion with "annual biomass". The same with perennial biomass (see table 2, etc.).
The abbreviation DD is introduced later (117 and then unnecessarily one more time – lines 206, 208) than it first appears in the text (fig.1). Then in the article names of sites “Danielov dom” (DD) and “Horný Smokovec” (HS) sometimes mentioned as full names, sometimes as abbreviation.
In table 1 authors mentioned Betula spp. as B. pendula and B. pubescens, but then in the text (275) gives as B. pendula only.
In the note to Table 1: “*** Other broadleaves were calculated by allometric relationship derived for Populus tremula”. Populus tremula is not broadleaved species, as well as Betula, Sorbus, Salix. So, better to use term «deciduous».
The first sentence in the «Results» (205-206) section is unclear (… species for either of transects of which 3 (min. 0, max. 8) were tree taxa), editing is required.
Line 211 - better «and» than «but»
Lines 235-237: “This means that individual plants identified within a plot did not have to be rooted inside the plot, instead the trees growing outside the plot simply enlarged their crowns that grew into the plot”. This conclusion is true only with coverage, but not frequency.
Table 3 is not readable - data «stuck together».
298-300 «Mean number of all species per plot, including tree species, did not change signifi-cantly over the studied period. This corresponds to common successional trajectories with rapid post-disturbance increase of plant richness and consequent gradual decline» These two sentences contradict each other. The first refers to the absence of change, and the second to a sharp increase immediately after the disturbances. The word “corresponds” is not good in that case.
331-334 Authors should not talk about the climate of one concrete year, it is more correct to use the word «weather» or «weather conditions».
346 «Graminoids are in general much better adapted to high light» - It is not good idea to reduce everything to light and the word “adapted” in this case does not reflect the essence because studied graminoid species are light-loving and receive benefit from the destruction of the tree layer. This is not an adaptation.
396-397 The competition for light was identified as the primary driver of herb layer biomass decrease. The influence of the tree layer is a much more complex phenomenon than just shading. The authors did not show the results of special studies that would allow making such statements.
Author Response
Comments and Suggestions for Authors:
it is necessary to correct the title "Short Term Dynamics of Vegetation Diversity and Aboveground Biomass of Picea abies Forests after Heavy Disturbance", because the authors consider only the consequences of the storm, and not all variants of severe desturbances.
The title has been corrected as: "Short Term Dynamics of Vegetation Diversity and Aboveground Biomass of Picea abies Forests after Heavy Windstorm Disturbance”
The authors in the methods section use both passive and active voice when presenting the material. It is better to use only active voice.
Yes, we have corrected the text according to the reviewerˈs recommendation.
Latin plant names are not always italicized (see 226, 227, 231 etc.).
We apologize for that. It was caused by the different Word formatting of the template. It should be correct in the revised version now.
There is confusion between annual biomass, biomass of annual species and biomass of annual parts of herbs. Annual biomass for example in the sentence (l 214) means the biomass that has grown over the year, but in the context it is the biomass of annual shoots. Authors should clearly mark "biomass of annual shoots" through the text (for example using an abbreviation - BAS) so that there is no confusion with "annual biomass". The same with perennial biomass (see table 2, etc.).
We were aware that finding the right term is important not to confuse the readers and keep the specificity. We appreciate the suggestion. Based on the reviewer´s suggestion, literature, and the fact that the terms refer to herb layer biomass. we changed the terms to „biomass of annual herb shoots“ and „biomass of perrenial herb shoots“. These terms and their abbreviations BAHS and BPHS were used uniformly throughout the entire manuscript.
The abbreviation DD is introduced later (117 and then unnecessarily one more time – lines 206, 208) than it first appears in the text (fig.1). Then in the article names of sites “Danielov dom” (DD) and “Horný Smokovec” (HS) sometimes mentioned as full names, sometimes as abbreviation.
We have corrected these details. The abbreviations are explained when they appear for the first time and full names are used just once where the abbreviations are introduced.
In table 1 authors mentioned Betula spp. as B. pendula and B. pubescens, but then in the text (275) gives as B. pendula only.
It is corrected in the revised version. The analysed data included just species Betula pendula.
In the note to Table 1: “*** Other broadleaves were calculated by allometric relationship derived for Populus tremula”. Populus tremula is not broadleaved species, as well as Betula, Sorbus, Salix. So, better to use term «deciduous».
We changed the word broadleaved to deciduous in the entire text.
The first sentence in the «Results» (205-206) section is unclear (… species for either of transects of which 3 (min. 0, max. 8) were tree taxa), editing is required.
We have corrected the sentence as follows:
Previous version:
Mean species richness recorded per 16 m2 plot was 17 (min. 9, max. 26) species for either of transects of which 3 (min. 0, max. 8) were tree taxa (Table 2).
Corrected version:
Mean species richness recorded per 16 m2 plot was 17 (min. 9, max. 26) for either of transects. On average, three species out of the total 17 were tree taxa (min. 0, max. 8) (Table 3).
Line 211 - better «and» than «but»
Done
Lines 235-237: “This means that individual plants identified within a plot did not have to be rooted inside the plot, instead the trees growing outside the plot simply enlarged their crowns that grew into the plot”. This conclusion is true only with coverage, but not frequency.
We do not agree and suppose that this is a small misunderstanding. The frequency was calculated from species presence at 25 plots located along transects. When the crown of the neighbouring tree (growing - rooted next to the plot) grew into the plot, it was recorded as a presence of that particular tree. Consequently, also frequency of that tree species within transect increased.
Table 3 is not readable - data «stuck together».
Corrected.
298-300 «Mean number of all species per plot, including tree species, did not change signifi-cantly over the studied period. This corresponds to common successional trajectories with rapid post-disturbance increase of plant richness and consequent gradual decline» These two sentences contradict each other. The first refers to the absence of change, and the second to a sharp increase immediately after the disturbances. The word “corresponds” is not good in that case.
The sentence was completely rewritten as follows:
Species richness of post-disturbance vegetation usually rapidly increases after disturbance event and later when competitive species take place it gradually declines.
331-334 Authors should not talk about the climate of one concrete year, it is more correct to use the word «weather» or «weather conditions».
Indeed. Thanks for the correction. We used both suggested terms.
346 «Graminoids are in general much better adapted to high light» - It is not good idea to reduce everything to light and the word “adapted” in this case does not reflect the essence because studied graminoid species are light-loving and receive benefit from the destruction of the tree layer. This is not an adaptation.
We accept also this comment and accordingly have changed also this part of the discussion.
Previous version:
Graminoids are in general much better adapted to high light levels [e.g. 22], and therefore the identified decrease of annual biomass along with the increasing tree cover and continuing succession could be assigned mostly to grasses.
Current version:
These grassy species certainly benefited from higher light levels after disturbance [e.g. 11, 22]. Thus, identified decrease of annual biomass along with the increasing tree cover and continuing succession could be assigned mostly to grasses.
396-397 The competition for light was identified as the primary driver of herb layer biomass decrease. The influence of the tree layer is a much more complex phenomenon than just shading. The authors did not show the results of special studies that would allow making such statements.
Accepted and rewritten.
Previous version:
The competition for light was identified as the primary driver of herb layer biomass decrease.
Corrected version:
The increasing tree cover had stronger impact on herb layer biomass than tree biomass, which emphasize the importance of light conditions on herb layer biomass.
The final comment: English language of the whole manuscript has been revised.

This manuscript is a resubmission of an earlier submission. The following is a list of the peer review reports and author responses from that submission.
Round 1
Reviewer 1 Report
Overall: Introduction is a bit erratic. It is just about 1 page, but loaded with several complex terms not described well. This random mentioning of complex terms was the most distracting throughout the manuscript. There were several mentions of forest ecosystem services, that were not explained well, and I found that to be very distracting when reading the objectives of the paper because they were not aligned well. I found some major flaws in these almost all the sections that requires major attention; I list some specifics them below. In many instances I found the writing to be poor, sentences incoherent and sloppy. It may help to coherently answer each question as listed at the end of the introduction because I found it tough to completely understand what the answer to each of your questions were at the end of the discussion. This major disconnect between the introduction and the discussion is perhaps the biggest weakness of this paper. After I read this paper I found myself wondering why it is essential that readers know about this study. That is something I did not pick up, and I don’t think the authors did a good enough job at telling the readers why their study is important.
Abstract:
14: give at least one example of a forest ecosystem service. As it reads, I am not sure why a reader should care about post-disturbance development of vegetation.
16: I don’t know what short-term is
18: species composition and tree biomass were measured at
19: is 12 years post wind disturbance considered short term? – qualify the statement to connect with line 16
27: connect 12 and 14 to these years in line 19.
29-30: changed after a short period of time and ..
32: line 14 needs to connect to line 32.
Introduction:
Line 48: “using local seed dispersal” doesn’t read well. Consider “by local seed dispersal”.
Line 55: “carbon stocks” requires more in the sentences leading up to it. The first mention of carbon stocks linked to a lack of post-forest disturbance and if you are referring to succession here, then keep the terms consistent.
Line 65: services such as …
Lines 66-80. Very loaded section which left me asking several questions. The random jargon in this section requires more attention or removal. From one moment to the next this introduction started talking about carbon sequestration, then jumps to successional ecology, then back to carbon sequestration. At this point I’m really unsure what this paper is about.
Line 81: This
Line 81: tree-herb? You referred to this as “ground vegetation” in the previous paragraph. Be consistent.
Line 88: im missing the implications of these questions. What will answering these questions help us better understand?
Results:
- Mean number shouldn’t be a range. Either a number, a fraction, decimal or whole number +- s.d.but not a range, unless rephrase and remove “mean”.
91: Consider combining to previous sentence and write, “of which 3 were tree species”
90-94: reference to “mean number of species” comes up twice. A quick glance at your objective #1 refers to species richness. Why not write species richness then?
95: insignificant is not statistical, unless its meant to be a synonym for “small”. Otherwise, you should use non-significant or not significant (as you do in the next sentence and line 103).
- Overuse of “rather” here and almost throughout manuscript.
- not sure what optimal conditions are. This might be better in the discussion where you can elaborate on it.
106: Bray-Curtis
117: move to discussion
119: whats the F-value? 180? 180,1 seems like the degrees of freedom (which are generally reported as a subscript e.g. F180,1= value, p<0.01 or in brackets F(180,1)= value, p<0.01.
122: pioneer tree species
132: affected
132-134: Methods.
137: a clear
Page 5. x-axis showing percent should stop at 100%
Figure 2 and 3 caption. Explain what the loess shades refer to. Captions should be standalone sentences that can explain the entire graph.
146: Table 4 seems like it should be Table 1. Number of trees (per what?)
Table 5 and 6 are mentioned early in the results section. It should be mentioned sequentially when referred. Consider re-organizing these tables in the way they are presented in the paper. Both can also very well be supplemental. Also, consider how important they are as Table 5 is referred to once in this paper, and Table 6 not at all. Im not quite sure what the point of Table 6 is when it is not referred to.
Discussion:
160: climatic climax is a century old concept that this paper is not explaining. Consider rephrasing and/or deleting “climax”.
161: pioneer tree species
162: also found.
162: study objects? Not sure what that is.
166: the tree layer
167: the contribution… the entire… “climax”. explain it, or delete it.
168: advantage of
166-169. Very confusing sentence.
- over time?
- cite some papers to support that. Also, no expectations were made clear in the introduction. So I’m not sure what you expected. Make it clear or rephrase.
171-173. confusing incoherent sentence.
177-179: italicize latin names
- low cover.
182-184. inconsistent use of spatial heterogeneity and beta diversity.
191: more important, “but only when predicting annual biomass”
196: latin names
202: which implies that
203 presence? Do you mean survive? Or persist?
206: “very serious” is subjective
210: also very subjective. Your study shows that “ground vegetation” regeneration is quite fast and highly biodiverse. I think you are referring mostly to carbon sequestration by forests – which are lost when stands are lost. Be clear, this does not come across as very objective and this has been mentioned several times in this study. Carbon sequestration flux is not an objective of this study.
211: one of those examples you could mention upfront.
211-213. Timber is only produced to be harvested. Rephrase the sentence and acknowledge the conundrum. If there are exceptions to timber production when not harvested (and can be used for CO2 sequestration forever), then tell the reader about that.
214: I wouldn’t describe that as dynamic growth. Maybe large growth? Remove “a”
231: first mention of NPP = write out in full or add NPP acronym to first mention of full term.
229-232. I’ve seen this too often in this paper. Random mention of complex terms that goes without explanation. Almost no mention of NPP and LAI prior to this. You need to explain these terms and why you are telling me about these recovery rates and time lines. Knowing what NPP and LAI is, is not enough. Why should a reader be interested in recovery rates of these metrics?
- no idea what the applied management approaches are
- Another random mention of ecosystem services, now with groups. These need to be explained earlier if they are to be vital to your study.
- Im missing the answers to your research questions- maybe they will show up in our conclusions?.
Methods:
Some major writing errors, I cant keep up. Read this section over and fix.
247: delete especially.
263: transect length of 292m which should be representative of a 130km damaged area. Of course sampling the entire 130km area is not possible, but please justify your plot dimensions and study design.
- write out square root transformation, don’t refer to an R function, non-R users do not know what that means.
- same for wilcox function.
Reviewer 2 Report
Overall comment:
This manuscript describes an interesting survey of vegetation change at two sites over two time-points in a post-disturbance forest ecosystem in the High Tatra Mountains. I thought it was well written, but the explanation of the logic behind the study, as well as the clarity of the aims and conclusions needs substantial work, in my opinion. Given the replication of sites (n = 2), and the lack of control sites to provide context (what is the natural variability through time in an undisturbed forest? Are the disturbed forests becoming more similar to the undisturbed control sites?) it is difficult to draw any conclusions beyond the effect of tree biomass on annual herbaceous biomass. It would have been great to be able to investigate the way that management interventions have influenced forest development, which would provide insights to support future decision making. A reference site(s) would provide information that could demonstrate the trajectory of change (other than that the trees are getting bigger through time). I agree with the authors that continuous post disturbance monitoring is vital, but I'm not sure how this study provides any new insights towards supporting that argument. The addition of reference suites in mature forests as well as an increase in site level replication to explore environmental drivers as well as proximity to undisturbed forests, would have provided the types of information needed to draw broader conclusions that might be useful for analogous forest systems. I know this type of comment isn't helpful, and is probably pretty infuriating, as the study has already been done, but I'm afraid I can't think of a way to improve the manuscript without more data.
Abstract:
Ln 12: This statement might need a bit more explanation: how has climate change resulted in increased disturbance of forests? Via fire? Storms? Also, minor edit: delete ‘the’ from ‘the climate change’
Ln 14: change to ‘an extreme windstorm’ because ‘the’ implies that the reader knows of this windstorm.
Ln 27: Change to: Annual biomass of *the* herb layer…
Introduction:
I would like to know how exactly the structure of forests with a history of disturbance affects their vulnerability to further disturbance? I presume this is something about how well established the root systems are? I know that fire affected forests of Australia become more flammable in the years immediately following a high-intensity fire, but I’m not sure how this theory would apply to wind-throw disturbances. This needs more explicit explanation.
In general, some more introduction to how biomass measurements can help predict forest structure and processes would be useful. I didn't quite understand the link. If you make this link very clear, it would really help with the overall clarity of the paper.
Aims and hypothesis: It’s not clear what you were expecting to find from this survey method. Consider a rethink of the aims and conclusions from the study. I can see how this survey might have implications for on-ground management of these forests in the post-disturbance period, but you haven’t articulated which aspects of the forest recovery might be warning-signs that require management intervention, or specifically which aspects of the post-disturbance trajectory you are particularly concerned about. I think the introduction needs to be re-written to make the aims and hypotheses clearer.
Results:
A minor suggestion: perhaps categorising the non-tree species according to their known traits or habitats could help to make more sense of the results. Are there certain species that only occur in mature/undisturbed forests, that present in the plots after x years. Are there certain ruderal/early successional species that start to drop out after x years. And then relate this to tree biomass. I just think this data hasn’t been explored with a clear plan for uncovering patterns or processes that might be informative for future forest management.
Coniferous vs broadleaf trees: surely these different trees would have different effects on the understorey? Would separating them in the analysis partition out more of the variation in understorey? Also, for readers outside of this system, perhaps grouping tree species into categories would be more informative.
Is rainfall variable between years? Could this explain some of the annual/perennial biomass variability?
Methods:
The survey method was: 2 very long (200m+) transects with 25 square 4 x 4 m plots placed 8 m apart. These are very small plots, presumably this small plot size will not be relevant once the forest develops further and trees are larger? Could you not have done a greater number of shorter transects – increase number of sites, and include fewer plots (including some larger plots with smaller plots nested within) along the transects? If you had good reason for this design, it just needs more explanation. Because it’s not clear from the way the data was analysed why you chose this design.
How might have elevation affected biomass/diversity?
Did you expect a difference between the two sites? They seem to have slightly different context (one closer to an intact forest than the other). Other differences?
Why were there no ‘control’ transect to show what an undisturbed forest contains? If you are tracking change through time, it would have been good to have a reference to provide context. Are these disturbed plots becoming more similar to a long-undisturbed plot through time? Or are they on a novel trajectory? And why might this be the case (climate? Rainfall? Feedbacks within the system – a tip into an alternative disturbance regime?) Without this information it is difficult to understand what the overall purpose of this study was. Other than documenting change in the forest between two time points. And if this is the aim, then surely more sites would be needed to really get a sense for the heterogeneity across the affected area.
Line 253 – 258: This information is very important and yet, I find this description of the two sites very confusing. I think you might need to explain what interventions have occurred at each site in more detail. I don’t know what ‘salvage logging’ entails exactly and you can’t assume that this is widely understood. It is not clear what the revegetation method consisted of, and which species were planted and at what density (also add an asterix for planted species in tables 5 and 6). Can this information be included in a table, include elevation, slope, mean annual rainfall etc etc. This will make it clearer how the two sites differ.